# Immune Sensing and Potential Immunotherapeutic Approaches to Control Chromoblastomycosis

**DOI:** 10.3390/jof7010003

**Published:** 2020-12-22

**Authors:** Leandro C. D. Breda, Isabela G. Menezes, Larissa N. M. Paulo, Sandro Rogério de Almeida

**Affiliations:** Departamento de Análises Clínicas e Toxicológicas, Faculdade de Ciências Farmacêuticas, Universidade de São Paulo, São Paulo 05508-000, Brazil; leandrobreda@usp.br (L.C.D.B.); belinhagodoy@hotmail.com (I.G.M.); larissamonteiropaulo@usp.br (L.N.M.P.)

**Keywords:** chromoblastomycosis, *Fonsecaea pedrosoi*, immunotherapy, treatment, immune response

## Abstract

Chromoblastomycosis (CBM) is a neglected, chronic, and progressive subcutaneous mycosis caused by different species of fungi from the Herpotrichiellaceae family. CBM disease is usually associated with agricultural activities, and its infection is characterized by verrucous, erythematous papules, and atrophic lesions on the upper and lower limbs, leading to social stigma and impacts on patients’ welfare. The economic aspect of disease treatment is another relevant issue. There is no specific treatment for CBM, and different anti-fungal drug associations are used to treat the patients. However, the long period of the disease and the high cost of the treatment lead to treatment interruption and, consequently, relapse of the disease. In previous years, great progress had been made in the comprehension of the CBM pathophysiology. In this review, we discuss the differences in the cell wall composition of conidia, hyphae, and muriform cells, with a particular focus on the activation of the host immune response. We also highlight the importance of studies about the host skin immunology in CBM. Finally, we explore different immunotherapeutic studies, highlighting the importance of these approaches for future treatment strategies for CBM.

## 1. Chromoblastomycosis—First Case Report

Chromoblastomycosis (CBM) is a chronic, subcutaneous, and progressive mycosis caused by fungi from the Herpotrichiellaceae family. The main etiological agents of chromoblastomycosis are *Fonsecaea pedrosoi*, *F. monophora*, *F. nubica*, *F. pugnacious*, *Phialophora verrucosa*, *Cladophialophora carrionii*, *Rhinocladiella aquaspersa*, *Exophiala spinifera*, *Aureobasidium pullulans*, and *Chaetomium funicol* [1,2]. Like almost all fungal infections (maybe except for candidiasis and aspergillosis), CBM is a neglected tropical disease, affecting mainly rural workers and low-income people. The first report of the disease dates back to 1911 when two Brazilian physicians observed ulcerative and nodular skin lesions (similar to the lesions found in cutaneous blastomycosis) in four patients from the rural area of the Minas Gerais state [3]. Skin biopsies showed a brownish fungal structure while a filamentous colony with velvety-cotton aspects was observed in the lesion culture. Although the skin lesions were similar to those observed in cutaneous blastomycosis, the biopsy and culture findings suggested a different fungal disease. Therefore, Dr. Pedroso and Dr. Gomes temporarily named the disease as “black blastomycosis” and shipped the patients’ samples to Paris, for a complete mycology analysis. However, due to a critical moment in the world’s history (World War I, 1914–1918), these findings were only published in 1920 [4]. During this 9-year gap, between the initial contact with those four patients and the publishing of the article, a German physician working in Brazil, Dr. Maximilliano Willibaldo Rudolph, published a related case in a German journal in 1914 of about six patients with lower limb lesions caused by a black fungus. At that time, Dr. Rudolph named the disease “figueira disease”, since the patients were farmworkers working strictly with fig crops [5]. In 1936, an Argentinian researcher, Dr. Pablo Negroni, classified the fungus observed in the Brazilian patients (by Dr. Pedroso and Dr. Gomes, and also by Dr. Rudolph) as belonging to the *Fonsecaea* genus and, in honor of Dr. Pedroso, named the species as *pedrosoi* [6]. Since Dr. Rudolph has published the first case report of chromoblastomycosis in 1914, he is considered the “father” of this disease. However, the scientific community currently gives credit for the discovery of the disease to Dr. Pedroso and Dr. Gomes as their reported case started in 1911, despite not being published until 1920.

Although the disease was first reported in 1911–1920, the term chromoblastomycosis was suggested only in 1922 [7]. However, this term was not a concept at that time, and different names have been used all around the world to describe this disease, e.g., figueira disease, black blastomycosis, new blastomycosis, verrucous dermatitis caused by *Phialophora verrucose*, Pedroso disease, Fonseca disease, and Lane and Medlar disease, among others. However, in 1992, the International Society for Human and Animal Mycology stated that the nomenclature chromoblastomycosis is the correct one to describe this disease [8].

## 2. Epidemiology

Although observed worldwide, CBM is a disease that is often associated with tropical and subtropical climates with the majority of cases occurring in Brazil [9,10], Madagascar [11], Mexico [12], China [13], and Australia [4]. Since it is not a notifiable disease, it is difficult to precisely know the incidence and prevalence of CBM. Surveys and case reports in the literature suggest a prevalence of 1:6800 in Madagascar and 1:196,000 in Brazil [4,14]. CBM is a disease associated with rural workers, so it has been suggested that the natural habitat of the fungus is the environment, such as the soil and in several species of plants. However, the isolation of the pathogenic species from the environment is difficult. A couple of studies have isolated the saprophytic species from their natural habitat, but the pathogenic species have only been isolated in a few cases. Recently, a Brazilian group isolated pathogenic species from insects like bees, ants, and termites [15]. A typical example of pathogenic species recovered from the environment was observed in the state of Maranhão, in Brazil. A CBM outbreak was observed in this region in the 1990s, and the disease was diagnosed in farmworkers, specifically those working with babassu (*Orbignya phalerata*) crops, an economically important resource for the local population in Maranhão [16,17].

CBM infection is associated with skin lesions caused by infected plants, plant debris, thorns, or with the contact of a previous open wound with contaminated soil and leaves. Lesions in the upper and lower limbs are the most frequent symptom in CBM patients, and these are the body parts that are usually in contact with soil and plant debris. However, some patients have unusual lesions on the face, cornea, brain, chest, and abdomen [10,18,19,20,21]. Recently, a case of chronic fungal meningitis caused by *F. pedrosoi* was reported in a patient from an endemic region for tuberculosis in India [22], demonstrating the high virulence capacity of this fungus to cause human disease. Similar to other fungal infections, CBM treatment is challenging since it is costly, of long length, has moderate efficacy, and can present several side effects.

## 3. Fonsecaea Pedrosoi Cell Wall

Although CBM has several etiologic agents, *Fonsecaea pedrosoi* is the most prevalent species (up to 90% of the cases) worldwide [23], and it is characterized as a dimorphic (or polymorphic?) fungus due to its different developing forms. The saprophytic form is represented by conidia and/or hyphae and mycelia, and the pathogenic form is represented by sclerotic bodies (also called muriform cells), which are found in the host tissue and linked to the chronicity of the disease [23]. These forms are morphologically different, and each one has specific features. While the conidia are rounded structures measuring around 3–6 microns in diameter, the hyphae present as elongated and septated structures with various sizes up to 100 microns [24]. The pathogenic form, called sclerotic bodies, is represented by rounded structures, ranging from 5 to 12 microns in diameter, and they are only observed in host tissue [25]. To date, little is known about their morphological differentiation characteristics. However, Dong and colleagues (2020) recently reported that the host immune system can influence the fungal morphology [26]. Using athymic animals, they observed that *F. pedrosoi* hyphae progressively changed into muriform bodies in skin tissue, followed by macrophage phagocytosis. However, no hyphae growing reversely from the muriform bodies was observed in these animals. The muriform bodies were able to reproduce in mouse tissue, leading to severe damage in the animal’s footpad. Meanwhile, animals treated with immunosuppressants (cyclophosphamide) showed hyphae growing reversely from the muriform bodies, without any inflammatory clearance [26]. This study may raise questions about CMB’s pathogenesis. For example, could the muriform cells be transcutaneously implanted into the host’s skin and so cause the disease? Or, could this form survive in the environment and be infectious? These are questions that are yet to be answered.

Besides the differences in size and shape, the conidia, hyphae, and muriform bodies also have different components in their cell walls. It is important to clarify the differences in the expression of cell wall components from all forms of *F. pedrosoi* to get a better understanding of the host’s innate immune system activation in CBM disease. These components, called pathogen-associated molecular patterns (PAMPs), are recognized by the Patter-recognition receptors (PRRs) present on our innate immune cells. The most common fungal PAMPs are β-glucans, chitin, and mannans, and their expression level presentation on fungal cell walls directly influences innate immune system activation or inhibition. The fungal cell wall is a heterogenic structure that can be modified with regard to its composition, depending on the species, morphology, growth cycle, and microenvironment [27]. The fungal wall cell is formed by two layers (inner and outer) [24], which are mainly composed of three major β-glucans components: β-(1,3)-glucans, chitin, and mannans [27]. The inner layer, a preserved structure amongst fungal species, is located near the plasmatic membrane while the outer layer is located on the most external region. Apart from the morphological structure, the characteristics of the cell wall composition of different forms of *F. pedrosoi* have been shown (Figure 1). Studies have usually compared the conidial and hyphal forms, but not the sclerotic bodies. Compounds of sialic acids, a family of monosaccharides, have been reported on the surface of *F. pedrosoi*, with N-acetylneuraminicacid predominating in the mycelium whereas N-glycolylneuraminicacid is predominant in conidia [28]. These saccharides have a structural role in this fungus and may be involved in cell interaction since they can either mask the recognition sites or serve as recognition determinants [29]. Other studies have demonstrated the presence of carbohydrates from glucose, mannose, galactofuranose, rhamnose, and glucosamine in the walls of both conidia and hyphae [30,31]. Nevertheless, each form has different amounts of these carbohydrates; for example, rhamnosis is predominant in the conidia and galactofuranosis is predominant in the hyphae [31]. Another important component of the fungal cell wall is chitin, a polysaccharide consisting of a long-chain polymer of N-acetylglycosamine (GlcNAc) that plays a role in fungal structure and differentiation [4]. Dong and colleagues reported a chitin-like component of *F. pedrosoi* muriform cells that is able to impair the immune recognition of β-glucans by dectin-1 as well as inhibiting Th17 development [32]. Mannose and GlcNAc residues with a capacity of adhesion are present on the surface of *F. pedrosoi* conidia, as shown previously by Limongi and colleagues [33]. Their involvement in the process of adhesion to epithelial cells suggests they are involved in host interactions.

Regarding lipids, a greater amount of this compound is present in the conidia than in the hyphae. Palmitic, stearic, and oleic acids are presented in both morphologies, while the arachidonic and linoleic acids are only observed in the conidia and hyphae, respectively [30,31]. Fungal ceramides consist of a carbohydrate unit bound to a ceramide (which is a lipid), also called glycosylceramides or ceramide monohexosides (CHM). Present on the fungal surface, they have an important role in the host immune response beyond fungal morphogenesis. Several studies have demonstrated the presence of antibodies against CMH in the serum of fungi-infected patients, confirming its antigenic properties [34]. Since CMH is related to fungal differentiation, the antibodies against cerebrosides have been used to inhibit the morphological transitions of *Cryptococcus neoformans* [35] and *Candida albicans* [36]. In 2004, Nimrichter and colleagues characterized the cerebroside from muriform cells in the mycelial and conidial forms of *F. pedrosoi* according to its structure, antigenicity, and biological functions [37]. The CHM structure of the conidia and mycelia was found to be the same, N-2′-hydroxyhexadecanoyl-1-β-D-glucopyranosyl-9-methyl-4,8-sphingadienine; however, the CHM of the muriform cells contains 9-methyl-4,8-sphingadienine, carrying an extra hydroxyl group. CMH is also associated with muriform cell differentiation and melanin deposition. The monoclonal antibody anti-ceramide inhibits conidia growth and improves its clearance by macrophages, while the sclerotic cell recognition only occurs where melanin is poorly expressed [37,38]. Melanin is a component of the outer layer of the melanized fungi, like *F. pedrosoi* and *Aspergillus fumigatus* [24]. The CBM agents have a high concentration of this pigment on the cell wall, which provides their typical features: dark brown or black stain [39]. Although three different pathways have been observed in melanin synthesis—the dihydroxyphenylalanine (L-DOPA) pathway, 1.8-dihydroxynaphthalene (DHN) pathway, and the L-tyrosine degradation pathway—the most important for CBM agents is the DHN pathway [40,41]. Melanin has been associated with *F. pedrosoi* virulence and immune evasion, and its high synthesis inhibits macrophage phagocytosis activity [42]. Moreover, due to the great surface/volume ratio, a high rate of melanin deposition is observed in muriform cells, contributing to the virulence and immune evasion of these particles and leading to the chronicity of the disease. Although carbohydrates and lipids are the main components of the fungal cell wall, the surface protein expression is of paramount importance for host immune response modulation; however, little is known about protein expression in CBM agents. The abundance and diversity of proteins of *Exophiala dermatidis*, one of the CBM species, were analyzed under temperature stress conditions [43]. A high temperature (45 °C) did not induce any stress responses at the protein level, suggesting that fungal adaptation occurs at elevated temperatures. Proteomics studies are essential to increase the understanding of the key proteins involved in infectious environments. In *A. fumigatus*, for example, RodA hydrophobin is an important protein to increase fungal survival, since it masks dectin-1-dependent host immune responses [44,45]. Besides, Voltersen and colleagues described CcpA, another conidial surface cell protein of *A. fumigatus* that is essential for virulence. Therefore, more CBM studies that evaluate the fungal proteome are necessary [46].

We believe that the different membranes and cell wall components may influence the host–pathogen interactions; however, how these specific components interact with the host immune system is still unclear. Our recent study demonstrated that TLR-2 is essential for conidia killing by neutrophils, but not for hyphae killing [47]. However, TLR-2 is known to interact with beta-glucan, a component expressed in the two morphotypes of *F. pedrosoi*. Therefore, it is still not clear yet whether beta-glucan is hidden in the hyphae membrane, avoiding host immune system activation, or if the size of the hyphae can interfere with the host immune system activation. These data show that the host immune system activation is not only based on the fungal cell wall component, but also on the structure of the cell wall and the size of the particle. Therefore, a deep knowledge of both fungal structures and immunological responses is crucial for the development of an effective therapeutic approach against CBM agents (Figure 1).

## 4. Skin Immune System, an Overview

The innate immune system is composed of physical (skin and mucosa), chemical (pH, peptides, and proteins), and biological (leukocytes) barriers, while the adaptive immune system is composed of lymphoid lineage cells (T and B lymphocytes) and antibodies. Since CBM is a deep subcutaneous mycosis, skin histology and immunology are of paramount importance for host defense. The skin is the largest organ in the body and is composed of three distinct layers: the epidermis, dermis, and hypodermis (or subcutaneous fat tissue) (Figure 2). The epidermis is the outermost layer of the skin and has four different strata: the stratum corneum, the granulosum, the spinosum, and the basale [48,49]. Keratinocytes are the most abundant cell in the epidermis (comprising around 90% of the cells) and it is found in the four strata of the epidermis. The keratinocyte layer is known to immunologists and mycologists as an important physical barrier of the skin that protects the host against environmental threats. However, keratinocytes have several other functions that may be very important to the host’s defense against CBM infection, preventing the settle and the spread of the disease. Although keratinocytes are not immune cells, they have been described as expressing several PRRs on their surface [50]. The PRR–PAMPs interaction activates keratinocytes, leading to the production and secretion of antimicrobial peptides (AMPs, such as defensins and cathelicidins) [51,52,53], complement system proteins [54], and several proinflammatory cytokines [55,56]. It has been shown that keratinocytes kill *Candida albicans* through a mannose-binding receptor [57], and they are also important cells in the host’s skin defense against *Malassezia* [58], *Trichophyton* [59], and *Paracoccidioides* [60]. The keratinocytes of the epidermis’ outermost layer (stratum corneum) are called corneocytes and are characterized as dead-flatted anuclear keratinocytes. The main function of corneocytes is to protect the viable cells of the lower layer (granulosum stratum), being an important physical barrier to the body, known as our first defense line against foreign threats [48,61]. Below the corneum stratum is the stratum granulosum, which is composed mainly of live keratinocytes and their secreted products, AMP, complement system proteins, and cytokines. Therefore, the stratum granulosum is the first important chemical barrier in the skin. The two deepest epidermis strata are the only ones to be composed of resident leukocytes. The spinosum stratum is composed of γδ T cells, Tissue resident memory T-cells (T_RM_ cells), Langerhans cells (LCs), and Dendritic epidermal T-cells (DETCs—only found in mice) [55,62,63,64], while the basale stratum is composed mainly of melanocytes, LCs, Merkel cells, and T-lymphocytes [56,61,65]. Therefore, these two deepest strata of the epidermis are important biological barriers of the skin that help to avoid the settlement and spread of an infection.

The skin layer under the epidermis is the dermis, and this contains the most abundant resident leukocyte population in the skin layers, including dendritic cells (DCs), macrophages (MOs), mast cells (MCs), b-lymphocytes, γδ T-cells, and innate lymphoid cells (ILCs) [56]. Other immune cells, such as neutrophils, αβ T-cells, and natural killer cells (NKs), are present in small numbers. Besides the immune cells, the dermis is also composed of collagen, elastin fibers, and an extensive network of blood and lymph vessels. The many blood vessels are crucial for the recruitment of immune cells in response to an infection or skin lesion [55,65], and the lymphatic system is essential for draining the DCs to the lymph nodes, where the adaptive immune system can be activated. Below the dermis is located the deepest layer of the skin tissue: the hypodermis or subcutaneous fat tissue. This layer is mainly constituted of fibrocytes and adipocytes (or adipose tissue). The adipose tissue contains the innate immune cells (including neutrophils, macrophages, dendritic cells, eosinophils, NKs, and ILCs) and adaptive immune cells (including T and B lymphocytes). The balance of these cells and the cytokines presented in the adipose tissue can be very distinct in lean and obese people [66].

The most frequent histological findings in CBM disease are hyperkeratosis (thickness of the stratum corneum, often due to abnormal expression of keratin), pseudo-epitheliomatous epidermal hyperplasia, pyogranulomatous reactions, and irregular acanthosis (thickness of the stratum spinosum and basale) or areas of atrophy [67,68,69]. The dermis presents a granulomatous inflammatory infiltrate composed of mononuclear cells (histocytes, lymphocytes, and plasmocytes), epithelioid cells, giant cells (Langerhans), and polymorphonuclear cells. It was shown that neutrophilic micro-abscesses may be present inside of granulomas [68]. Therefore, the skin tissue is not only a physical barrier against foreign pathogens but also a complex and rich active-immune organ. Unfortunately, the skin’s immune system is frequently neglected in CBM disease studies, probably due to the complexity of this organ. This review will detail how the host immune system works in CBM, so that we can discuss and suggest future potential immunotherapeutic approaches to control the CBM disease.

## 5. Immune Response in CBM

Since the first case report, in the early twentieth century, researchers worldwide have been working to understand how our immune system works against CBM infection. After more than 100 years of the discovery, a lot has been done to understand the immune response against this fungus. In the sections below, we discuss the most important data on the CBM immune response in the literature.

### 5.1. Adaptive Immune Response in CBM

One of the first published works in the literature observed that murine models of CBM showed disseminated disease, with live fungi in organs like the brain, liver, lungs, heart, spleen, and kidneys. The authors suggested that the cellular response is more protective to the host than the humoral response [70]. The significance of T lymphocytes in the control of CBM disease has been observed in experiments using nude animals (T-cell deficient mice). In the absence of T-cells, a severe disease was observed in these animals. However, after an adoptive T-cell transfer, the infected animals showed complete remission from the disease [71]. The following studies demonstrated that the CD4^+^ T lymphocytes are the subpopulation responsible for the control of the disease. The absence of CD8^+^ T lymphocytes did not interfere with the severity of the disease, at least in murine CBM [72]. Since CD4^+^ lymphocytes (especially Th1 subset) are important to IFN-γ production, the role of CD4^+^ lymphocytes was also observed in humans, whereby patients with mild symptoms of CBM showed higher levels of IFN-γ and lower levels of IL-10 than patients with severe symptoms of CBM [73,74]. Recently, in vitro experiments showed that *F. pedrosoi* drives CD4^+^ Th17 cell differentiation [32,75], while an increase in the IL-17 level was observed in skin lesions of murine CBM [76]. IL17 was also showed to be important in the early stage of the disease. Animals that had IL-17 blocked by anti-IL-17 injections showed a greater fungal burden during the early stages of the disease [77]. IL-17 is an important cytokine that is related to AMP release in the skin and fungal infections [78,79,80], helping with the killing of extracellular pathogens, such as hyphae. Although related to the polarization of anti-inflammatory macrophages (M2 macrophages) [81,82], IL-17 is an important cytokine for neutrophil migration. It is related to the increases in MIP-2 and KC, important chemokines in neutrophil attraction [83,84]. Therefore, the blockage of IL-17 could interfere with AMP release and neutrophil attraction to the infection site, leading to an increase in the fungal burden; thus, neutrophils are important in *F. pedrosoi* conidia and hyphae killing (as discussed below). However, the roles of the Th17 cell population and IL-17 in the control of CBM disease still remain unclear. 

Although the cellular response is important for CBM control, the role of the humoral response in the disease is not fully determined yet. In the 1980s, one group demonstrated the capacity of *F. pedrosoi*-specific IgG antibodies to decrease 50% to 60% of fungal growth in vitro [85]. Later, a specific antibody to *F. pedrosoi* melanin was isolated and demonstrated to inhibit fungal growth and increase human neutrophil phagocytosis, as well as reactive oxygen species (ROS) production [86]. In 2010, Machado and colleagues observed that B-cell-deficient mice (Xid animals) had more severe CBM than wild-type animals, suggesting that the humoral response is essential for the control of the disease [87]. However, the patients’ samples showed conflicting results about the importance of the humoral response in CBM disease control. Esterre and colleagues first observed that IgG levels were directly related to disease severity [88], but this finding was not observed by Gimenez and colleagues [73]. Therefore, the actual consensus is that the CD4^+^ Th1 lymphocyte response is associated with better control of the disease, while the CD4^+^ Th2 lymphocytes are related to poor control and higher disease severity [89] (Figure 3). The importance of a humoral response in CBM disease needs deeper and specific studies.

### 5.2. Innate Immune Response in CBM

Most studies of CBM’s immune response have focused on the innate immune response. In 1984, Torinuki and colleagues observed the activation of the alternative pathway of the complement system over *F. pedrosoi* [90]. Thereafter, it was identified that melanin, present in a high concentration in the *F. pedrosoi* cell wall, was triggering the activation of the alternative pathways of the complement system [91,92]. The complement system’s activation seems to be important for the opsonization (C3b) and chemotaxis process (C3a and C5a), which drives the innate immune response towards fungal clearance. However, due to the *F. pedrosoi* cell wall thickness, it is highly unlikely that the complement system acts to directly eliminate *F. pedrosoi* through formation of the membrane attack complex. Another important innate immune component is the macrophage. In vitro experiments have demonstrated that *F. pedrosoi* conidia ingested by peritoneal resident macrophages can survive and grow inside the macrophages, transforming into hyphae particles and leading to macrophage death through mechanical disruption of the cytoplasmic membrane [42]. Previous activation with IFN-γ will enable macrophages to block fungal growth inside it, acting as a Trojan horse and carrying live fungal particles around the host [93]. The inability of macrophages to kill ingested conidia is due to the capacity of *F. pedrosoi* to block nitric oxide production in macrophages, even after IFN-γ activation [94,95]. Therefore, although the levels of IFN-γ are related to a mild form of the disease, the mechanism of IFN-γ to act against CBM agents is still unclear. Unlike macrophages, neutrophils show a high in vitro microbicidal activity against *F. pedrosoi* [47,96]. In 2011, in an analysis of skin biopsies from mice infected with three different forms of *F. pedrosoi* (conidia, sclerotic bodies, and hyphae), Machado and colleagues observed dead fragments or injured fungal particles in neutrophilic areas, while intact or particles similar to the live particles were observed in regions of macrophage cells. These findings suggest that neutrophils are important for *F. pedrosoi* killing, not only in vitro but also in vivo. The authors also suggested that neutrophils release Neutrophil Extracellular Traps (NETs) to kill hyphae particles since they found an area of degenerative neutrophils surrounding hyphae particles in the skin biopsies [97], findings also observed by Dr. Ogawa and colleagues later on [98]. Recently, our group demonstrated that neutrophils kill conidia and hyphae in vitro in a distinct manner. Whilst conidia killing relies on phagocytosis and ROS production, neutrophils kill hyphae particles through ROS-independent Neutrophil Extracellular Trap (NET) release [47]. These findings, together with the increase of IL-17 in skin lesions of CBM patients, suggest that neutrophil attraction and activation are the most important roles of IL-17 in CBM infection, leading to *F. pedrosoi* killing and avoiding the spread of the disease.

Another important type of innate immune cell is the DC. These cells are important in the defense against several fungal infections and have a high migratory capacity, cytokine production, and expression of co-stimulatory molecules upon activation, which makes them the perfect bridge between the innate and adaptive immune systems (for a review of DCs and fungal infection, see [99]). Unlike macrophages and neutrophils, the role of DCs in CBM reported in the literature is sometimes contradictory. One of the studies using DCs obtained from the mouse skin (Langerhans cells) showed that the *F. pedrosoi* conidia block T-lymphocyte activation by inhibiting the CD40 and B7.2 expression in Langerhans cells in vitro [100]. However, our group demonstrated that monocyte-derived dendritic cells from CBM patients positively modulated HLA-DR and CD86 after conidia stimulation in vitro, leading to CD4^+^ Th1 lymphocyte differentiation and proliferation [101]. Since these studies focused on DCs in different parts of the body, more studies are needed to improve the understanding of the DCs’ role in CBM disease.

### 5.3. Pattern Recognition Receptors (PRRs)

The PRRs–PAMPs interaction is one of the most important and most-studied parts of the host’s strategy in pathogen recognition. The PRRs are located in the cell surface or in intracellular vacuoles, and several pathogen-associated molecular patterns (PAMPS) have been recognized. They are present in large amounts in APCs and innate immune cells, but it can also be found in other immune cells, such as B and T lymphocytes, and also in non-immune cells, such as keratinocytes [102,103], and melanocytes [104], present in our skin. Two of the most studied PRR classes are Toll-like receptors (TLRs) and C-type lectin receptors (CLRs).

Although a variety of studies have demonstrated the importance of these two classes of receptors in different fungal infections [45,105,106,107,108,109,110], little is known about their roles in *F. pedrosoi* recognition. Recently, a new receptor of DHN-melanin was discovered by Stappers and colleagues and named the Melanin sensing C-type Lectin receptor (MelLec). This discovery might be important for understanding the host immune response in CBM infection, where muriform cells have a higher concentration of melanin in the cell wall. It is the infection morphotype of the fungus, leading to the chronicity of the disease [111] (Figure 3). Another recent study demonstrated that Dectin-1 [112] and Dectin-2 [75] are important in *F. pedrosoi* recognition, stimulating the Th1 and Th17 responses, respectively. However, CBM agents bind to MINCLE, a new discovery receptor of the CLR family that inhibits and evades the host immune system. Fungal interaction with MINCLE leads to intracellular signaling that blocks IL-12 production (which was stimulated by Dectin-1 activation), inhibiting the Th1 differentiation of CD4+ lymphocytes [113] (Figure 3). This inhibition occurs through the IRF1 degradation after fungal recognition by the MINCLE receptor [112], leading to Th2 differentiation favoring the pathogen survival and infection spread. The MINCLE activation also inhibits Th17 differentiation, but its inhibition mechanism is not clear yet. Recently, Castro and colleagues demonstrated that *F. pedrosoi* conidia and hyphae activate the NLRP3 inflammasome, an important intracellular receptor belonging to the NOD-like receptor family (Figure 3). NLRP3 activation leads to IL-1β production in DCs and macrophages; however, IL-1β was not essential for the of control CBM disease, since no differences were observed in in vitro and in vivo experiments using NLRP3 KO animals [114]. Although several studies have focused on understanding the role of TLRs in fungal infections, almost no studies can be found in the literature regarding the roles of these receptors in CBM infection. A recent study showed an increase in tlr-2 and tlr-4 gene expression by macrophages infected with *F. pedrosoi* sclerotic bodies, but not with conidia. Later, our group demonstrated that TLR-2 and TLR-4 are essential receptors for *F. pedrosoi* conidia recognition and killing by murine neutrophils [47] (Figure 3). Although these receptors are important for conidia phagocytosis and ROS production, they are dispensable for hyphae killing through NET release. 

Confirming the in vitro findings, TLR-2KO, TLR-4KO, and Myd88-KO animals infected with *F. pedrosoi* conidia showed poor prognoses compared with wild-type animals, demonstrating that the TLRs are an essential class of PRRs for the recognition and control of CBM agents [47,113]. Although fungal recognition by TLRs has proved to be important in the early stages of the disease, it seems that *F. pedrosoi* has the ability to hide from the host’s immune system. Sousa and colleagues (2011) observed that animals pretreated with LPS had a better immune response in CBM, leading to a mild disease followed by its cure. This work demonstrated that *F. pedrosoi* has the ability to hide or to trick the host immune system by being invisible to the host immune system, leading to the settlement, spread, and chronicity of the disease [113]. This is possible probably due to the cell wall complexity of this fungus, which insufficiently stimulates PRRs, leading to an inappropriate host immune response that favors the fungal survival and disease spread. The authors demonstrated that, after exogenous stimulation of the innate immune system by LPS-TLR4 or imiquimoid-TLR9 activation, the host immune response was restored, providing disease resolution. This costimulatory response also requires the CLR Mincle and signaling via the Syk/CARD9 pathway. LPS is a well-known macromolecule that is able to induce a strong immune response in mammalian cells via the TLR4 pathway, promoting the secretion of proinflammatory cytokines, including tumor necrosis factor-α (TNF-α) and interleukin-1 (IL-1) [115]. Therefore, the treatment of patients with LPS can boost the immune response against fungi and improve their clearance; however, its toxicity impairs its medicinal use [116]. Similar to LPS, imiquimod is a cytokine inducer and an innate immune booster that works through the agonistic activity of TLR7, inducing cell recruitment and activation with IFN-gamma production [117]. Imiquimod is a well-characterized compound. It is FDA-approved and the topical 5% cream is already being manufactured and commercialized [117].

Based on these findings, Souza and colleagues (2014) later reported a new treatment for CBM patients, including a topical cream with the TLR agonist, which was demonstrated to lead to significant improvement of the skin lesion (as discussed below) [118].

## 6. Immunotherapy Approaches

As mentioned above, CBM treatment is a therapeutic challenge. Although several methods and antimicrobials agents have been used in attempts to cure or control the disease, the high rate of relapses and the low cure rate, observed especially in severe and chronic cases, make CBM treatment a real challenge. Currently, CBM agents are not effectively controlled by a specific drug or by drug association, so the search for alternative methods, particularly those based on immune modulation, is necessary. Nevertheless, a defect in innate recognition results in the failure to mount robust inflammatory responses, causing infection susceptibility and chronicity [113]. Immunotherapy aim to stimulate or restore the ability of the immune system to fight against an infection, overcoming the host immune impairment. Currently, antifungal immunotherapies may target not only the host—such as in adoptive cell transfer, the cytokine therapies, the modulatory mAbs/AMPs, and the vaccine approaches—but also the pathogens, such as the monoclonal antibody (mAb) and the antimicrobial peptide (AMPs) treatments [119].

Studies aimed at enhancing host immune responses against CBM agents are being developed (Table 1). Some immunostimulant agents, like imiquimod and glucan, are capable of overcoming this defect, enhancing the inflammatory responses, which results in clearance of the pathogen and resolution of the infection. Imiquimod is a synthetic compound that stimulates both the innate and adaptive immune pathways through Toll-like receptor 7 (TLR7). This synthetic compound has potent antiviral, antitumor, and immunoregulatory activities. It was approved by the Food and Drug Administration (FDA) for the treatment of external genital and perianal warts (condylomata acuminata) as well as for the treatment of actinic keratosis and superficial basal cell carcinoma [120]. Recently, the role of exogenous administration of a TLR7 agonist (imiquimod) in stimulating the inflammatory response mediated by Syk/CARD9 through a Mydd88-dependent pathway was demonstrated [113]. According to this study, four patients were treated with topical imiquimod (5%) five times per week, with or without antifungal drugs. The treatment was effective and provided not only clinical improvement but also a cure [118]. Belda and colleagues also reported a successful regression of lesions in patients treated with imiquimod as a monotherapy. It is interesting that both reports observed a transitory exacerbation of the verrucous and infiltrative characteristics of the lesion that preceded its gradual evolution to healing [121]. This finding highlights the ability of imiquimod to restore the host’s immune system, inducing a beneficial inflammatory response against the CBM agent.

In addition to imiquimod, treatment with Glucan is an immunotherapy that has been suggested for CBM treatment. Glucan is a carbohydrate that is a compound of the cell wall of several fungi, and there is evidence that this molecule has a positive effect on the treatment of diseases since it can stimulate innate immune responses. Clinical improvement of a CBM patient treated with glucan was reported by Azevedo and colleagues [122]. The authors reported significant lesion regression in a severe form of CBM after Glucan and itraconazole (ITZ) treatment in a patient with a 3-year history of unsuccessful treatment with only antifungal drugs. Glucan administration once a week with ITZ improved the cellular immune response of the patient and might be an alternative/adjuvant to treat this disease. Similar results were also demonstrated in *Candida* spp. [123] and *Aspergillus* spp. [124] infection.

Another important strategy to prevent and cure fungal disease is the development of vaccines. Several studies have uncovered components of the fungal cell wall that stimulate protective antibody production, making them good candidates for vaccines. Although it is not clear whether the humoral immune response is protective in CBM, it is known that CBM agents induce a highly specific humoral response in people living in endemic areas [125]. Some studies using mAb against glucosylceramide (GlcCer) and melanin have demonstrated the great potential of this therapeutic approach in CBM treatment. In vitro experiments using Mab against fungal GlcCer showed the inhibition of *F. pedrosoi* conidial growth and an the enhancement of the antifungal functions of murine macrophages [37]. However, this mAb is not completely effective against muriform cells because they were only able to recognize them at cell wall regions without melanization [38]. mAb treatment against melanin was also shown to be protective for the host. Melanin is a pigment present in a high concentration in chromoblastomycosis agents’ cell walls, and it is an important virulent factor that protects the fungi against destruction by immune cells [4,42]. The melanin of *Aspergillus* protects conidia from macrophage phagocytosis through Ca^2+^ sequestration, impairing cell homeostasis, and inhibiting the LC3-associated phagocytosis (LAP) pathway [126]. Due to the importance of this wall cell component in pathogen virulence, antibodies targeting *F. pedrosoi* melanin have been developed. The human antibody anti-melanin inhibits the in vitro growth of conidial and sclerotic cells [86]. In addition, soluble melanin improves the antifungal capacity of human neutrophils by enhancing phagocytosis and oxidative burst [86]. Although anti-melanin antibodies increase the phagocyte efficiency, some patients become close to achromic, either during the course of the disease or after the treatment [127]. This side effect suggests the occurrence of a cross-reaction with human melanocytes, leading to the development of vitiligo after immunotherapy.

**Table 1 jof-07-00003-t001:** Features of the reported chromoblastomycosis immunotherapeutic approaches.

Immunotherapy Class	Immunotherapy Agent	Mechanisms	Dose	Study	Protocol	Antifungal Association	Treatment Outcome
Immunostimulants	Imiquimod	Toll receptor 7 agonist	5% of imiquimod	Human	Five times a week (topical) for 6 to 19 months	Monotherapy	Lesions healed after 6 to 7 months [120]
					Five times a week (topical) for 6 to 17 months	Monotherapy or ITZ and/or TERB	Improvement in clinical aspects [117]
	β1,3 glucan	T cell proliferation and IFN-γ production	5 mg	Human	Once a week (IM) for 2 years	ITZ	Resolution of the majority of lesions [121]
DNA Vaccine	DNA-hsp65 vaccine	Reduction in NO production	9 mg	Mice	Once each 15 days (IM) during 15 or 30 days	Monotherapy or ITZ and/or AMB	Healed injuries and eliminated *F. pedrosoi* from lesions [127]
Monoclonal antibody (Mab) therapy	Mab anti-GlcCer	Fungistatic and fungicidal activities	NA	In vitro	NA	NA	Inhibition of *F. pedrosoi* growth in vitro and enhanced antifungal activity of murine macrophages [37]
Purified antibodies	Purified antibodies anti-Melanin	Fungicidal activities	NA	in vitro	NA	NA	Inhibition of *F. pedrosoi* growth in vitro [86]

Abbreviations: AMB, amphotericin B; CBM, chromoblastomycosis; IM, intramuscular; ITZ, itraconazole; Mab, monoclonal antibodies; NA, not applied; TERB, terbinafine; GlcCer, glucosylceramides.

Immunostimulants or vaccines may improve the immune system, leading to host protection, and can be an adjuvant to antifungal drugs, which could reduce the treatment time and costs, increasing its success. Currently, there is no licensed vaccine for the prevention or treatment of human CBM or any mycosis. A conventional vaccine may be a live-attenuated or inactivated microorganism or a subunit that is able to stimulate the adaptive immune system, leading to host immune memory. A DNA-hsp65 vaccine either in association or not with intralesional administration of itraconazole and amphotericin B or not was effective for treating experimental CBM [128]. With regard to vaccine subunits, several antigens from fungal pathogens have been described, such as proteins, carbohydrates and polysaccharides, and lipids, and the peptide vaccine has emerged as a promising advance in immunotherapy. The advantages of peptide-based vaccines include easy production on a large scale with high purity; the possibility of being stored at room temperature, since it may be freeze-dried; the target specificity and induction of a proper immune response; and the reduced risk of side effects [129]. The main weakness of this vaccine, on the other hand, is the low immunogenicity; however, it can be improved through the development of better adjuvant-based delivery systems [130] or bioengineered molecules with more than two peptides [131]. Dendritic cells (DCs), as described above, play an important role in both innate and adaptive immune responses, and because of their efficiency in activating T cells and inducting memory-like responses in vivo [132], DCs may be a promising approach for vaccination. Mice protectively immunized with *C. neoformans* displayed a DC polarization phenotype inducing high IFN-γ and enhanced pro-inflammatory cytokine responses upon a subsequent challenge. In 2011, Magalhães and colleagues showed the therapeutic or prophylactic effects of DCs primed with peptide 10 (P10), a gp43 peptide derived from *Paracoccidioides brasiliensis*, in mice challenged with a virulent isolate, improving the lung fungal clearance [133]. Likewise, the administration of P10-primed DCs to an immunosuppressed mouse was also effective in combating *P. brasiliensis* infection. Until now, neither therapeutic DCs nor immunogenic peptides towards CBM treatment have been reported, but there are some research groups involved in this field for the development of peptide vaccines and DC pulses.

## 7. Final Considerations

Chromoblastomycosis is a neglected mycosis that poses a challenge regarding the treatment of patients with chronic disease and severe injuries, many of whom remain with clinical signs for months to years, even after prolonged antifungal treatment, which may lead to secondary infections and sometimes work inability. Knowledge of CBM’s chronicity, occurring as a result of innate recognition failure by the immune system [113], provides strong evidence that the immunotherapies applied to modulate the host’s immune responses, to overcome this failure, are potential strategies for an effective CBM therapy. Advances in monoclonal antibodies and peptide vaccine studies are a promising approach for this goal, since these compounds are highly specific and may be protective through the stimulation of the pathogen phagocytosis or the induction of cytokine release with few side effects. Currently, imiquimod is the most effective immunotherapy for *F. pedrosoi* infection amongst those studied so far, acting as a good adjuvant in CBM therapy. Additionally, advances in the bioengineering field are future prospects for fungal infection, with CAR-T cell therapy being successfully reported for the treatment of some malignancies. Although it is widely studied in cancer—due to targeting a tumor-associated antigen—these T-cells carrying a chimeric antigen receptor (CAR-T cell) may be engineered to any other antigen, including fungal antigens, providing alternative tools for targeting fungal pathogens. Despite the effort focused on understanding the pathogenic mechanism of the CBM agents and their antigenic properties, the achievement of a vaccine or immunotherapy approach against chromoblastomycosis still faces challenges. Knowledge of the host’s specific fungal proteome is essential for the identification of new vaccine candidates. Furthermore, a deep elucidation of immune sensing during host–fungal interactions is also fundamental to provide useful data for immunotherapy approaches.

## Figures and Tables

**Figure 1 jof-07-00003-f001:**
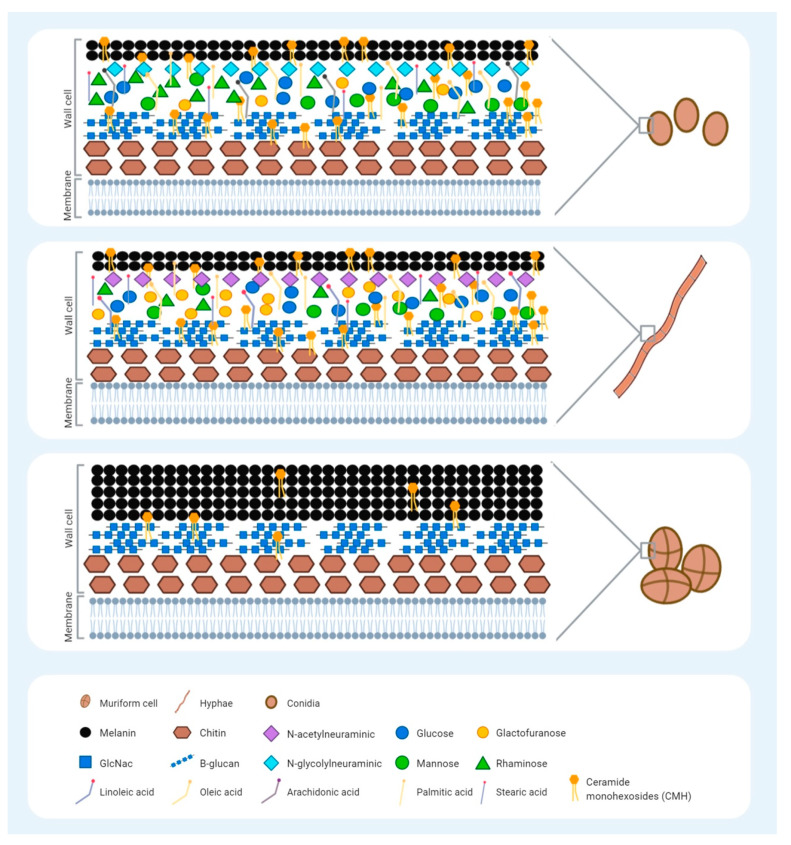
Schematic representation of the *Fonsecaea pedrosoi* cell wall, highlighting the reported molecules in each morphological form: conidia, hyphae, and muriform cells. The fungal cell wall is a heterogenic structure that is important for activation of the host innate immune system during infection through the pathogen-associated molecular patterns (PAMPs) recognized by the Pattern-recognition receptors (PRRs) present on innate immune cells. The morphological forms of *F. pedrosoi* have similarities but also differences with regard to the type and amount of the molecules present. No difference in chitin and B-glucan expression have been reported for the three different forms of *F. pedrosoi*. Conidia predominantly express sialic acid N-glycolylneuraminicacid, whereas hyphae express N-acetylneuraminicacid. Carbohydrates, such as glucose, mannose, and glucosamine, are similarly deposited on conidia and hyphae surfaces; however, conidia have greater expression of rhamnosis, while hyphae have greater galactofuranosis expression. Conidia also present a greater amount of lipids than hyphae, and the expression of the arachidonic fatty acids seems to be exclusive to the conidia form. The expression of linoleic fatty acids is exclusive to the hyphae, while the palmitic, stearic, and oleic fatty acids are present in both morphologies. Ceramide monohexosides (CHM) consist of a carbohydrate unit bound to a ceramide with antigenic properties. Presenting on the surface of all three *F. pedrosoi* morphologies, it is related to fungal differentiation. Melanin is a pigment present in the outer layer of all forms of *F. pedrosoi*, which provides their typical dark brown or black stain. A higher deposition of melanin is observed in muriform cells that block the binding sites of the anti-CHM antibodies, contributing to a greater virulence and immune evasion in these cells.

**Figure 2 jof-07-00003-f002:**
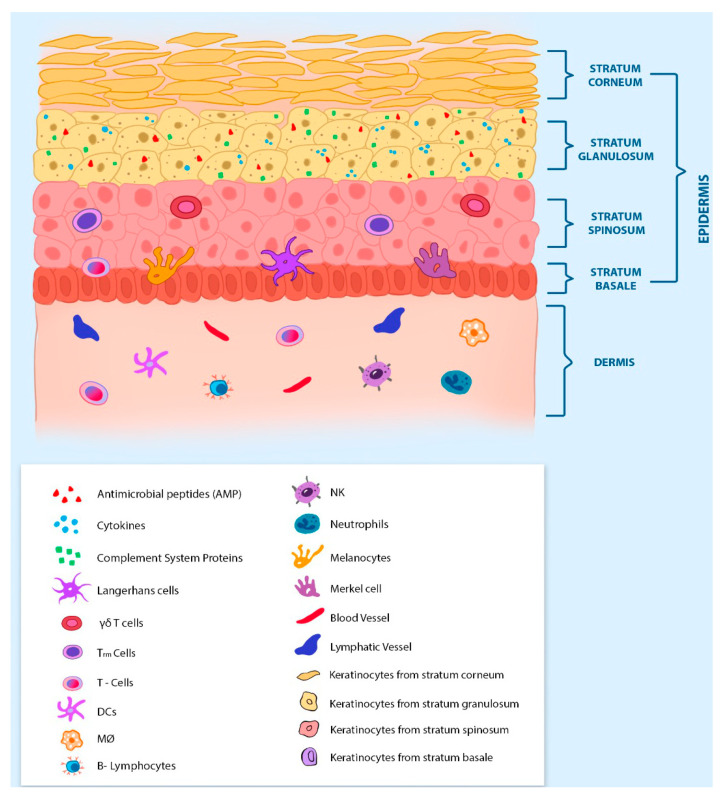
**Immune system and histology of the skin:** The epidermis and dermis are the two of the most important layers in the skin’s immunity. The epidermis is the outermost layer and is composed of approximately 90% keratinocytes, presenting four specific stratums. The stratum corneum is composed of terminally differentiated keratinocytes, also called corneocytes. The stratum granulosum is composed of granular keratinocytes. This is the first epidermis stratum with components of the immune system, such as cytokines, antimicrobial peptides, and proteins from the complement system. Below the stratum granulosum is the stratum spinous. The keratinocytes of the spinous stratum have cytoplasmic spinous processes that extend to meet the processes of the adjacent cells. This is the first epidermis stratum with immune cells like T-cells, γδ T-cells, and Tissue-resident memory T-cells (TRM cells). Besides the immune cells, the interface between the stratum spinous and stratum basale is composed of melanocytes, Merkel cells, and Langerhans cells. The stratum basale is the innermost layer of the epidermis and it is in close contact with the dermis, which has the greatest variety of immune cells. The dermis has an abundant vascular system, with blood and lymphatic vessels. The most prevalent types of cells in the dermis are T and B cells, DCs, NKs, neutrophils, and macrophages. These two skin layers represent an important physical and biologic barrier to the host. Although most of the studies about CBM disease neglect the skin’s immunity, an understanding of this immune organ is of paramount importance for the control and treatment of the disease.

**Figure 3 jof-07-00003-f003:**
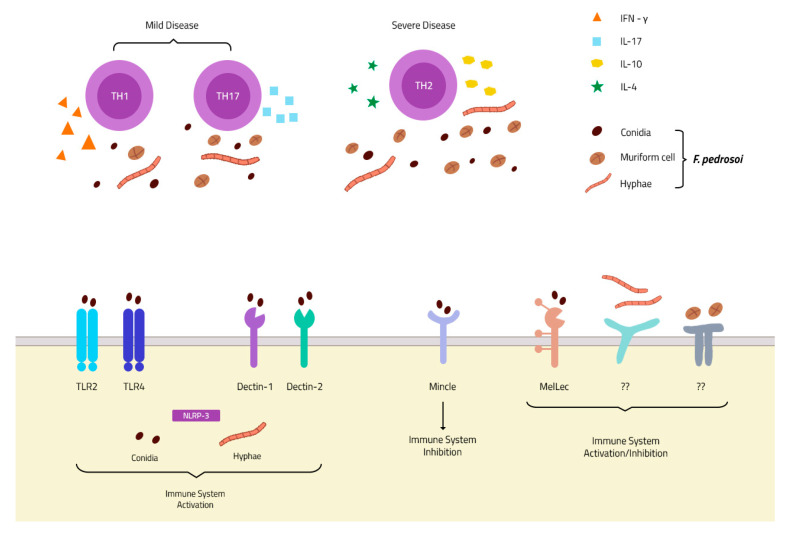
**PRRs and immune response in chromoblastomycosis (CBM) infection:** Membrane receptors of the TLR and CLR families described in *F. pedrosoi* conidia recognition. To date, only one receptor is described in hyphae recognition—the intracellular NLRP-3 inflammasome—and none is described in muriform cell recognition (**Bottom**). The Th1 and Th17 response is associated with mild disease, and an increase in IFN-γ and IL-17 secretion, while a Th2 response is related to severe disease with increases in IL-4 and IL-10 secretion and fungal spread (**Top**).

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
