# Peer review of "Immune Sensing and Potential Immunotherapeutic Approaches to Control Chromoblastomycosis"

_jof, 2020, doi:10.3390/jof7010003_

Round 1

Reviewer 1 Report

Very nice review. Some English spelling correction needed, and check sentences like 'researchers have been neglected with the skin immune response in CBM', I dont know what that means. Also Trojas horse (not horse Trojan), and similar small things. The graphic are very nice but display the simplest part of the story; would it be possible to make an overarching diagram of all cells, receptors and cytokines involved? That would help understanding a lot. In stead, the history part could be shorter, as this has already been described in several other papers.

Author Response

Please find enclosed the revised manuscript entitled “Immune sensing and potential immunotherapeutic approaches to control chromoblastomycosis”, which we re-submit to publication.

All suggestions recommended by the referees were analyzed and corrected in the new version of the manuscript, as follows:

Reviewer #1:

  1. Some English spelling correction needed, and check sentences like 'researchers have been neglected with the skin immune response in CBM', I dont know what that means. Also Trojas horse (not horse Trojan), and similar small things.

Answer 1: We appreciate the reviewer’s comments and suggestions. The manuscript was now revised by “MDPI English Editing Service” to correct all spelling, grammar, and typos

  1. The graphic are very nice but display the simplest part of the story; would it be possible to make an overarching diagram of all cells, receptors and cytokines involved? That would help understanding a lot.

Answer 2: We appreciate the reviewer's comments and suggestions. A new figure was included in the manuscript (Fig. 3) to summarize all the receptors, cytokines and Th cells involved in CBM infection

  1. In stead, the history part could be shorter, as this has already been described in several other papers.

Answer 3: We appreciate the reviewer’s comments and suggestions. We did some amendments in the history part, leaving it more concise as suggested.

Reviewer 2 Report

The review by Breda and colleagues explores the clinical and experimental aspects of chromoblastomycosis. As mentioned this is a neglected tropical disease which has not garnered much attention. However, progress has been made in understanding the pathogenesis and immune response to this fungus. Efforts are being made to create a vaccine. While this review is timely, there are multiple issues with it.

  1. The English needs substantial improvement in order for readers to comprehend exactly what the authors mean.
  2. The authors describe PAMPs and PRRs twice-once in section 3 and again in section 4. No need to belabor the point.
  3. The review contains a textbook-like description of the immune system and that is totally unnecessary. Rather, the authors should just keep a focus on what is important to this specific disease. Statements like “The immune system is composed of organs…” are totally unnecessary.
  4. The authors describe the content of the cell wall (section 3) and then allude to the fact that different morphotypes would engage the immune system differently. That may be true but then never go on to describe and define what components of the cell wall are engaging the innate immune system and how differences in carbohydrate or protein would sway the immune system. This leads to an unsatisfying result.
  5. The authors discuss adaptive immunity they spend a considerable amount of effort on IL-17, but then merely speculate that IL-17 works by promoting neutrophil influx. But neutrophils cannot eat hyphae so how does that work? They neglect other scenarios such that IL-17 promotes defensin release, and IL-17 has been shown to arm macrophages. More depth here is needed. And not much is mentioned about interferon gamma-again an important cytokine in arming cells that kill fungi.
  6. The authors never touch on the fact that this is a skin disease. Therefore, this fungus has adapted to survive the cooler temperatures on the skin. What is the reason for this and how does this shape our thinking about disease pathogenesis. If you contrast with other fungi that have skin manifestations such as blastomycosis or coccidioidomycosis, we know they survive in both warm and cooler temperatures. Why is Fonseca different? And how is the immune system recognition different at the cooler temperatures than that found in visceral organs.
  7. There are good descriptions of studies but little in the way of synthesis of what the compendium of studies means. This needs to be reworked. What is the significance of LPS pretreatment and how would that differ from imiquimod?

Author Response

Please find enclosed the revised manuscript entitled “Immune sensing and potential immunotherapeutic approaches to control chromoblastomycosis”, which we re-submit to publication.

All suggestions recommended by the referees were analyzed and corrected in the new version of the manuscript, as follows:

Reviewer #2:

  1. The English needs substantial improvement in order for readers to comprehend exactly what the authors mean.

Answer 1: We appreciate the reviewer’s comment and suggestions. The manuscript was now revised by “MDPI English Editing Service” to correct all spelling, grammar, and typos.

  1. The authors describe PAMPs and PRRs twice-once in section 3 and again in section 4. No need to belabor the point.

Answer 2: Thank you for pointing out this flaw. We remove the PAMP and PRR description in section 4

  1. The review contains a textbook-like description of the immune system and that is totally unnecessary. Rather, the authors should just keep a focus on what is important to this specific disease. Statements like “The immune system is composed of organs…” are totally unnecessary.

Answer 3: We appreciate the reviewers’ comments. We did some amendments in the text, especially in section 4, improving the manuscript, and removing the unnecessary paragraphs and descriptions.

  1. The authors describe the content of the cell wall (section 3) and then allude to the fact that different morphotypes would engage the immune system differently. That may be true but then never go on to describe and define what components of the cell wall are engaging the innate immune system and how differences in carbohydrate or protein would sway the immune system. This leads to an unsatisfying result.

Answer 4: We appreciate the reviewers’ comments. Several studies describe the specific interaction of fungal cell wall components and PRRs (PMID 29522806 and 25420452). This review aims to discuss the difference in cell wall components between the different morphotypes of F. pedrosoi and the PRRs related to the control of CBM infection (as described in section 5.3). Making a bridge between each fungal cell wall component and its specific host immune system activation/inhibition is a complex subject, and not clear yet (at least in CBM disease). TLR-2, for example, recognizes beta-glucan that is present in F. pedrosoi conidia and hyphae. However, our study demonstrated that TLR-2 is essential to conidia but not hyphae killing (at least by neutrophils). This data shows that the host immune system activation does not occur only based on the fungal cell wall component. A specific component can be expressed by two different morphotypes of the fungus, however, it can be immunogenic in one form but not in another one, once that this component can be hidden and cannot be recognized by the host immune system in one of these two forms. We did some amendments in line 198-209 to make it clear to the readers and also added some new information regarding the MelLec receptor in lines 457-462.

  1. The authors discuss adaptive immunity they spend a considerable amount of effort on IL-17, but then merely speculate that IL-17 works by promoting neutrophil influx. But neutrophils cannot eat hyphae so how does that work? They neglect other scenarios such that IL-17 promotes defensin release, and IL-17 has been shown to arm macrophages. More depth here is needed. And not much is mentioned about interferon gamma-again an important cytokine in arming cells that kill fungi.

Answer 5: Regarding the role of IL-17 in CBM infection, we included the information about defensins and macrophages in Line 372-380. However, we believe that the greatest contribution of IL-17 in CBM infection is due to the neutrophil attractions since we have demonstrated that neutrophils can kill conidia and hyphae of F. pedrosoi. Regarding the hyphae killing by neutrophils, we discussed it in line 426-429. Our findings showed that neutrophils release NETs to kill hyphae in a ROS-independent manner. The IFN-gamma in CBM infection was discussed in lines 411-416. Although this is an important cytokine in several fungal diseases, in vitro experiments demonstrated that macrophages pre-activated with IFN-gamma showed only fungistatic but not fungicidal activity. Therefore, the IFN-gamma role in CBM is still unclear.

  1. The authors never touch on the fact that this is a skin disease. Therefore, this fungus has adapted to survive the cooler temperatures on the skin. What is the reason for this and how does this shape our thinking about disease pathogenesis. If you contrast with other fungi that have skin manifestations such as blastomycosis or coccidioidomycosis, we know they survive in both warm and cooler temperatures. Why is Fonseca different? And how is the immune system recognition different at the cooler temperatures than that found in visceral organs.

Answer 6: We appreciate the reviewers' comments. As mentioned in the abstract in section 4, CBM is a skin disease but rare are the studies focusing on skin immunology to understand this disease. The normal human skin temperature varies between 33.5 and 36.9 °C, though the skin's temperature is lower over the nose and higher over muscles and active organs. Studies evaluating the skin immune defense against the pathogen in a temperature-dependent manner are scarce. In a study about the innate defense of airway cells against rhinovirus, the authors showed changes in the virus-host interaction, leading to a reduced innate immune response by infected airway cells and a relative decrease of IFN and IFN- stimulated genes (ISGs) at cool temperature (33°C) (PMID: 25561542). Harper et al (PMID: 29760065) also observed that the NF-κB dynamics are affected by increased and decreased temperatures. Pathogenic species of Fonsecaea had optimum development at 33°C and its maximum growth temperature is 37°C (PMID: 29062304). Given that this fungal infects the skin, where the temperature is lower, and the immune response could be impaired in cool temperatures, we believe that F. pedrosoi has a favorable characteristic to grow/develop and persist in the skin. However, this suggestion is mere speculation because there are no reports that confirm this.

  1. There are good descriptions of studies but little in the way of synthesis of what the compendium of studies means. This needs to be reworked. What is the significance of LPS pretreatment and how would that differ from imiquimod?

Answer 7: We appreciate the reviewer’s comments. We made some amendments to clarify the LPS/Imiquimod treatment to our readers. We added in line 506-519 the following statement: “The authors demonstrated that after exogenous stimulation of the innate immune system by LPS-TLR4 or Imiquimoid-TLR9 activation, the host immune response was restored providing disease’s resolution. Besides, this costimulatory response also requires the CLR Mincle and signaling via the Syk/CARD9 pathway. LPS is a well-known macromolecule able to induce a strong immune response in mammalian cells via TLR4 pathway, promoting proinflammatory cytokines secretion - including tumor necrosis factor-α (TNF- α) and interleukin-1 (IL-1) (PMID: 29099761). Therefore, the patient’s treatment using LPS can boost the immune response against fungal and improve its clearance, however its toxicity impairs its medicinal use (PMID: 18289078). Similar to LPS, Imiquimod is a cytokine inducer and an innate immune booster through the agonistic activity of TLR7 inducing cell recruitment and activation with IFN-gamma production (vidal and alomar, 2008). Imiquimod is a well-characterized compound, FDA approved and the topical cream 5% is already manufactured and commercialized (DOI: 10.1586/17469872.3.2.151).”

Round 2

Reviewer 2 Report

This is somewhat improved. The English still needs work although the authors state that they used MDPI's english service. Subjects and verbs don't always align. Again, there is no synthesis of the findings and but rather just reporting. They have inserted some of my suggestions but have integrated them well into the discussion. For example, they state that IL-17 may increase defensins, but without indicating that mouse neutrophils do not possess alpha defensins. Overall, this is a modest contribution to the field but a contribution nonetheless.